# Long-period variability in ice-dammed glacier outburst floods due to evolving catchment geometry

Amy Jenson[1,2], Jason M. Amundson[2], Jonathan Kingslake[3], and Eran Hood[2]

[1]Department of Mathematical Sciences, Montana State University, Bozeman, Montana 59717, USA
[2]Department of Natural Sciences, University of Alaska Southeast, Juneau, Alaska 99801, USA
[3]Lamont Doherty Earth Observatory, Columbia University, Palisades, New York 10964, USA

**Correspondence:** Amy Jenson (amyjenson@montana.edu)

**Abstract.** We combine a glacier outburst flood model with a glacier flow model to investigate decadal to centennial variations in outburst floods originating from ice-dammed marginal basins. Marginal basins can form due to the retreat and detachment of tributary glaciers, a process that often results in remnant ice being left behind. The remnant ice, which can act like an ice shelf or break apart into a pack of icebergs, limits a basin's water storage capacity but also exerts pressure on the underlying water and promotes drainage. We find that during glacier retreat there is a strong, nearly linear relationship between flood water volume and peak discharge for individual basins, despite large changes in glacier and remnant ice volumes that are expected to impact flood hydrographs. Consequently, peak discharge increases over time as long as there is remnant ice remaining in a basin and peak discharge begins to decrease once a basin becomes ice free. Thus similar size outburst floods can occur at very different stages of glacier retreat. We also find that the temporal variability in outburst flood magnitude depends on how the floods initiate. Basins that connect to the subglacial hydrological system only after reaching flotation depth yield greater long-term variability in outburst floods than basins that are continuously connected to the subglacial hydrological system (and therefore release floods that initiate before reaching flotation depth). Our results highlight the importance of improving our understanding of both changes in basin geometry and outburst flood initiation mechanisms in order to better assess outburst flood hazards and their impacts on landscape and ecosystem evolution.

## 1  Introduction

Glacier outburst floods (also referred to as jökulhlaups) are sudden releases of water from ice-dammed or moraine-dammed lakes. There has been a recent increase in the size and number of glacial lakes due to deglaciation (e.g. Clague et al., 2012; Shugar et al., 2020; Mölg et al., 2021), raising concerns about the hazards that these lakes pose to downstream communities and infrastructure. More accurate estimates of flood magnitude and timing may help mitigate risk in areas where these hazards exist (e.g. Vincent et al., 2010; Werder et al., 2010). In addition, outburst floods cause semi-regular but short-lived perturbations to downstream ecosystems by rapidly changing sediment and nutrient concentrations and proglacial water temperatures (e.g., Neal, 2007; Kjeldsen et al., 2014; Meerhoff et al., 2019). The largest of these floods create major erosional features during glacial periods (e.g., Larsen and Lamb, 2016; Keisling et al., 2020); smaller, more frequent outburst floods are also important in driving landscape change (e.g., Russell et al., 2006; Cook et al., 2018; Carrivick and Tweed, 2019). Here, motivated by

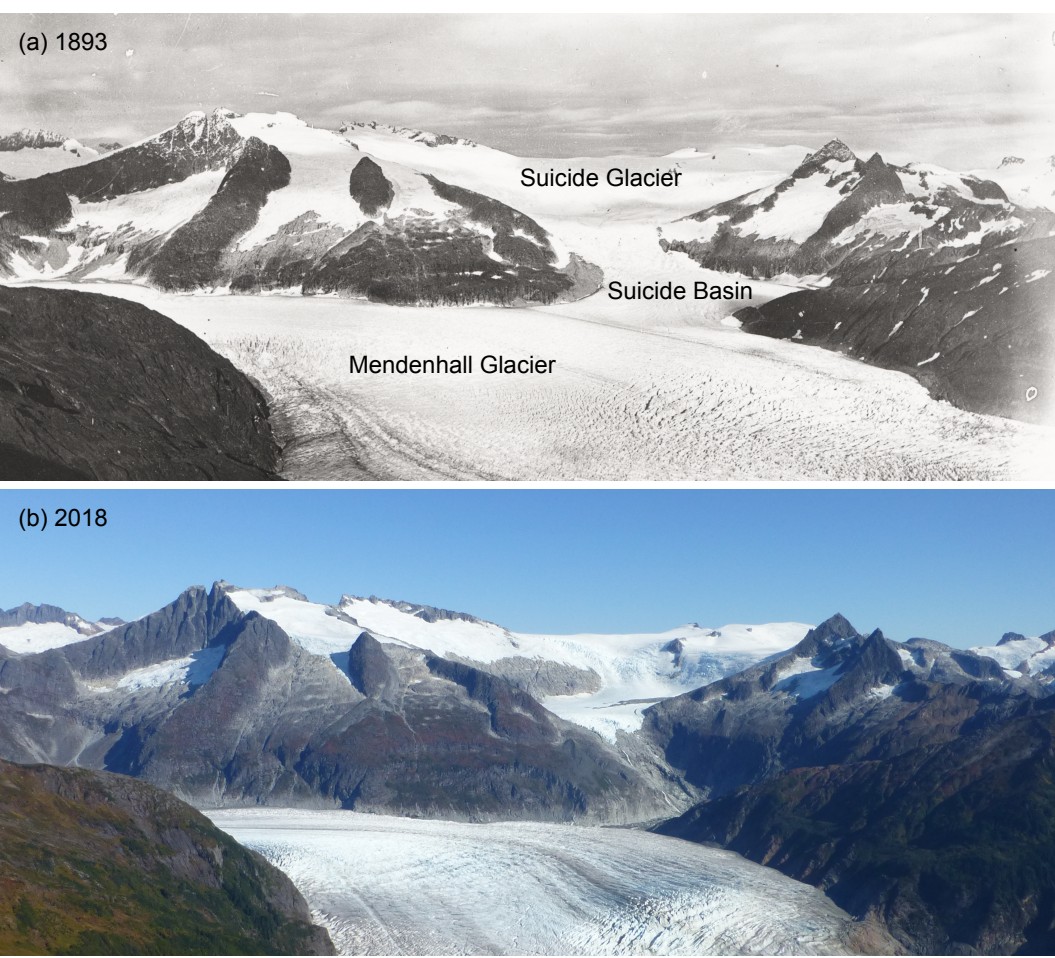

**Figure 1.** Repeat photos of Mendenhall Glacier, Alaska, taken in (a) 1893 (Ogilvie, 1893) and (b) 2018 (courtesy of C. Kienholz). Suicide Basin formed in the early 2000s when Suicide Glacier detached from Mendenhall Glacier. Annual outburst floods now originate from Suicide Basin, which contains remnant ice from Suicide Glacier.

observations from Mendenhall Glacier, Alaska, we focus on glacier outburst floods from ice-dammed marginal basins, which form following the thinning, detachment, and retreat of tributary glaciers and often contain remnant ice left behind during deglaciation (e.g., Capps et al., 2010; Kingslake and Ng, 2013a; Kienholz et al., 2020) (Fig. 1).

The theory of ice-dammed outburst floods is based on the consideration of mass, momentum, and energy balances of water flowing through the subglacial drainage system (e.g., Rothlisberger, 1972; Nye, 1976; Fowler, 1999; Kingslake, 2013; Kessler and Anderson, 2004; Stubblefield et al., 2019; Schoof, 2020). Many outburst flood models are based on the assumption of a circular or semicircular channel (e.g., Nye, 1976; Fowler, 1999; Kingslake, 2013); others allow for more complex drainage configurations such as multiple lakes in a connected hydraulic system (e.g., Stubblefield et al., 2019) or a system of linked cavities (e.g., Kessler and Anderson, 2004; Schoof, 2020), which may be more accurate at modeling outburst events early

in the melt season. In these models, once a flood initiates, the water begins to drain through an existing subglacial drainage
system. The energy dissipated in the flowing water causes the conduit(s) to grow and the discharge to increase until the peak
discharge is reached and the basin has drained. A positive feedback loop between discharge, melt rates, and conduit area
results in flood hydrographs that rise quasi-exponentially and then rapidly drop once the basin is empty (or nearly empty)
(Nye, 1976). The mechanics of flood initiation are less understood. One proposed mechanism is that a basin begins to drain
when the water pressure equals the overburden pressure of the ice dam, which occurs at the lake level referred to as flotation
depth (Thorarinsson, 1953). When an outburst flood initiates due to the basin reaching flotation depth, water floats the ice
dam and flows beneath the ice: (1) forming a channel, (2) enlarging an existing channel, or (3) propagating a subglacial sheet
of water toward the terminus (e.g., Flower et al., 2004). There are also many occurrences of outburst floods initiating prior
to a basin reaching flotation depth (e.g., Bjornsson, 1992) or alternatively exceeding flotation depth (e.g., Huss et al., 2007;
Kienholz et al., 2020). Several studies have also considered the possibility that marginal basins remain continuously connected
to the subglacial and englacial hydrological systems and that drainage onset is dictated by the interplay between the water depth
in the basin relative to the ice dam height, the hydraulic gradient in the vicinity of the basin, and the state of the hydrological
system (e.g., Kessler and Anderson, 2004; Kingslake, 2015; Bigelow et al., 2020; Schoof, 2020). Due to a poor understanding
of drainage onset, the timing and magnitude of outburst floods are difficult to predict (e.g., Ng and Björnsson, 2003; Kingslake
and Ng, 2013b).

Outburst flood theory dictates that flood characteristics, such as event timing and peak discharge, depend on glacier and
basin geometry, both of which evolve as glaciers advance or retreat. Consequently, outburst floods can be viewed as semi-
periodic disturbances to glaciated landscapes that switch on/off and evolve in response to climate change. We are motivated by
a desire to understand the evolving hazard of outburst floods as well as the impacts of these extreme events on landscape and
ecosystem evolution. Thus, our work complements efforts to understand long-term variations in glacier runoff during glacial
recession (e.g., Milner et al., 2017; Huss and Hock, 2018). In situ observations of outburst floods from individual glaciers over
multiple years or decades are limited to a few sites. Due to a lack of observations, no previous work has tried to develop a
theoretical understanding of the impact that glacier retreat has on outburst flood hydrographs. We address this problem with
a one-way coupled glacier-basin-outburst flood model and focus on quantifying the long-period variability in outburst floods
that arise due to changes in catchment geometry. Our primary objective is to investigate changes in outburst flood hydrographs
as a glacier retreats by exploring different basin geometries and flood onset mechanisms. In addition we account for remnant
ice left behind in a basin, which reduces the storage capacity of water in the basin but also acts like a gravity piston that pushes
water out of a basin. We do not attempt to address the significant year-to-year variability in outburst flood hydrographs that has
been observed at some glaciers (e.g., Huss et al., 2007; Neal, 2007; Kienholz et al., 2020); in this light our modeling efforts
should be viewed as an attempt to quantify the potential for a given glacierized catchment to produce outburst floods.

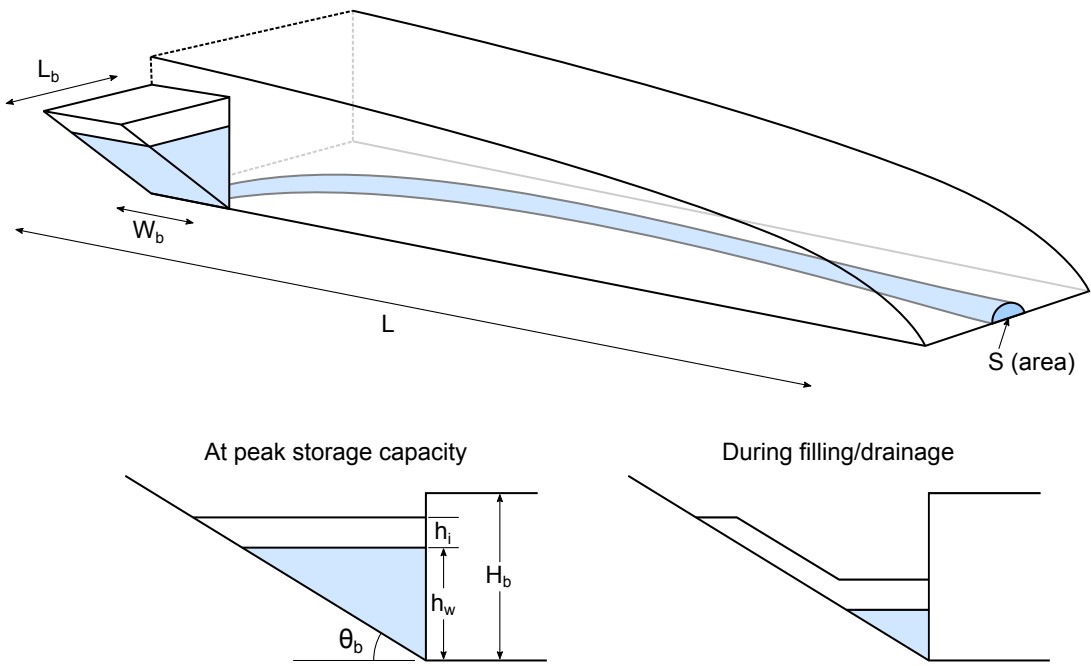

**Figure 2.** Model schematic illustrating the glacier and basin geometry (for a wedge-shaped basin).

## 2 Model description

We build on the outburst flood modeling work of Nye (1976), Fowler (1999), and Kingslake (2013) by accounting for changes in glacier and basin geometry (Fig. 2), both of which are expected to affect the magnitude and duration of outburst floods. We first use an idealized glacier flow model to quantify changes in glacier geometry, ice dam thickness, and the amount of remnant, floating ice in a basin as a glacier retreats. For each year of the glacier flow model we extract the glacier geometry and remnant ice volume, which we then feed into the glacier outburst flood model. In the following subsections we describe the outburst flood model, the hypsometry and evolution of the marginal basin, and the glacier flow model. A list of model variables is included in Table 1.

### 2.1 Outburst flood model

#### 2.1.1 Channel hydrology

The outburst flood model consists of four coupled equations that conserve mass, momentum, and energy as water flows from a marginal basin and through a semi-circular conduit to the glacier terminus, assumed to be open to the atmosphere (Nye, 1976; Fowler, 1999). The ice dam seal is assumed to be located immediately adjacent to the basin. The cross-sectional area of the conduit, $S$, evolves by melting and creep closure, and consequently discharge $Q$, effective pressure $N$ (ice-overburden minus water pressure), and melt rate $\dot{m}$ (expressed as mass per unit length per unit time) vary temporally and spatially. We define the

densities of ice and water as $\rho_i = 917$ kg m$^{-3}$ and $\rho_w = 1000$ kg m$^{-3}$, gravitational acceleration as $g$, and the latent heat of fusion as $L_f = 3.34 \times 10^5$ J kg$^{-1}$ (Cuffey and Paterson, 2010). Following Fowler (1999), we use the basic hydraulic gradient $\psi = \rho_w g \sin\theta - \partial P_i / \partial s$, where $\theta$ is the conduit slope (assumed to equal the bed slope), $P_i = \rho_i g H$ is the ice pressure, $H$ is the glacier thickness, and $s$ is the along-flow coordinate parallel to the bed. The conduit length $L$, glacier thickness, and glacier thickness gradient evolve as the glacier thins and retreats (Section 2.2).

The assumption that the channel walls enlarge by melt and shrink due to creep closure results in an expression for the rate of change of conduit area given by

$$\frac{\partial S}{\partial t} = \frac{\dot{m}}{\rho_i} - KSN^n, \tag{1}$$

where $K = 2An^{-n}$ (Evatt, 2015) and $A = 2.4 \times 10^{-24}$ Pa$^{-3}$ s$^{-1}$ and $n = 3$ are the flow law parameter and exponent in Glen's Flow Law (assuming temperate ice). Assuming pressurized flow, mass conservation dictates that the rate of change of conduit

area is also related to the spatial gradient in discharge, the production of meltwater, and additional water input to the conduit, such that

$$\frac{\partial S}{\partial t} + \frac{\partial Q}{\partial s} = \frac{\dot{m}}{\rho_w} + M, \tag{2}$$

where $M$ represents additional water flux supplied to the conduit per unit length. We prescribe a small value of $M = 10^{-5}$ m$^2$ s$^{-1}$ to ensure that the conduit always remains open (Fowler, 1999). We use Manning's equation to describe conservation of mo-

mentum, yielding an expression relating the discharge and conduit area to the basic hydraulic gradient and effective pressure,

$$\psi + \frac{\partial N}{\partial s} = f \rho_w g \frac{Q|Q|}{S^{8/3}}, \tag{3}$$

where $f = (2(\pi+2)^2 \pi^{-1})^{3/2} n'$ is a friction factor with $n' = 0.1$ m$^{1/3}$s the hydraulic roughness. Finally, conservation of energy requires that

$$\dot{m}L_f = Q\left(\psi + \frac{\partial N}{\partial s}\right). \tag{4}$$

Two boundary conditions are required to solve this system of equations. We set the effective pressure at the terminus equal to 0. At the basin outlet, the effective pressure is

$$N_b = \rho_i g H_b - (\rho_w g h_w + \rho_i g h_i), \tag{5}$$

where $H_b$ and $h_w$ are the glacier thickness and water depth at the basin outlet and $h_i$ is the thickness of floating ice in the basin.

Variations in water level are related to the basin hypsometry and discharge into and out of the basin, as described in Section 2.1.2 (see Eq. 10). In addition, the ice dam height and floating ice thickness both vary during glacier recession (Section 2.2).

### 2.1.2    Basin hypsometry and evolution

We assume that the ice-dammed basin has an idealized hypsometry that can be described by

$$A_b(z_b) = a z_b^{p-1}, \tag{6}$$

**Table 1.** List of model parameters. Values of constants are specified in brackets.

| Variable | Description |
| --- | --- |
| $\rho_i, \rho_w$ | densities of ice [917 kg m$^{-3}$] and water [1000 kg m$^{-3}$] |
| $g$ | gravitational acceleration [9.81 m s$^{-2}$] |
| $L_f$ | latent heat of fusion [$3.34 \times 10^5$ J kg$^{-1}$] |
| $A, n$ | ice flow law parameter [$2.4 \times 10^{-24}$ Pa$^{-3}$ s$^{-1}$] and exponent [3] |
| $K$ | ice flow parameter for conduit closure [$1.78 \times 10^{-25}$ Pa$^{-3}$ s$^{-1}$] |
| $f, n'$ | friction factor [0.066 m$^{-2/3}$ s$^2$] and hydraulic roughness [0.1 m$^{1/3}$] |
| $x, s, z, z_b$ | horizontal, bed-parallel, and vertical coordinates and elevation relative to ice dam base |
| $\theta, L, S, \dot{m}$ | conduit slope, length, cross-sectional area, and melt rate |
| $Q, Q_{in}, Q_b$ | discharge along the conduit, discharge into the basin, and discharge from the basin |
| $M$ | water flux to the conduit per unit length |
| $P_i, N, \psi$ | ice overburden pressure, effective pressure, and basic hydraulic gradient |
| $h_w, h_i$ | basin water depth and floating ice thickness |
| $S_0, h_{w,0}$ | initial cross sectional area and initial basin water depth |
| $H_b, N_b$ | ice dam thickness and effective pressure |
| $A_b$ | mapview area of the basin |
| $a, p$ | coefficient and exponent that describe basin hypsometry |
| $W_b, L_b, \theta_b$ | basin width, length, and bed slope |
| $V_i, V_w$ | volumes of ice and water in the basin |
| $V_s$ | basin storage capacity (volume of water when basin is at flotation depth) |
| $H, h_s, W, U$ | glacier thickness, surface elevation, width, and depth- and width-averaged velocity |
| $U_b, U_c$ | ice velocity toward the basin and calving rate into the basin |
| $\tau, \tau_{max}$ | basal shear stress and maximum basal shear stress [$2.5 \times 10^5$ Pa] |
| $\nu$ | ice viscosity |
| $\dot{B}, \dot{B}_{max}, \dot{B}_b$ | width-averaged, maximum, and basin specific mass balance rates |
| ELA | equilibrium line altitude |

where $A_b$ is the mapview area of the basin at different elevations, $z_b$ is the elevation relative to the lowest point in the basin, and $a$ and $p$ are constants that describe the basin shape. For reference, $p = 1$, $p = 2$, and $p = 3$ describe box-, wedge-, and semicircular-cone-shaped basins, respectively. We define $W_b$, $L_b$, and $\theta_b$ as the basin width, length, and bed slope (Fig. 2). For a box-shaped basin $a = W_b L_b$, for a wedge-shaped basin $a = W_b \cot \theta_b$, and for a semicircular-cone-shaped basin $a = (\pi/2) \cot^2 \theta_b$.

The basin is assumed to be completely filled with ice at year 0, at which point the tributary glacier detaches from the trunk glacier and leaves behind remnant ice. Initially the remnant ice may be attached to the trunk glacier and act like a floating ice tongue, but ultimately it breaks into a pack of icebergs. We assume that the remnant ice thins at a rate given by the specific surface mass balance rate. Thus we neglect replenishment of ice into the basin via glacier flow or iceberg calving. We further assume that the remnant ice is sufficiently mobile and fractured to form a horizontal layer of thickness $h_i$ as the basin fills. We therefore assume that drainage proceeds quickly enough that the floating ice thickness does not change during the course of the outburst flood and consequently ice is stranded on the basin walls (see Fig. 2). The floating ice volume at time $t$ is given by

$$V_i = V_{i,0} + \int_{t_0}^{t} \dot{B}_b A_b(H_b) \, dt',$$
(7)

where subscript ,0 refers to initial conditions, $\dot{B}_b$ is the specific surface mass balance rate (see Section 2.2) at the basin's elevation, and we apply the mass balance rate to the surface of the remnant ice.

The volume of water stored in the basin $V_w$ for a given water depth is

$$V_w = \frac{a}{p} h_w^p.$$
(8)

Since $a$ and $p$ are constants for a given basin, the water volume in the basin can be expressed as

$$V_w = \left( \frac{h_w}{h_{w,0}} \right)^p V_{w,0}.$$
(9)

The volume fluxes of water entering and leaving the basin are $Q_{in}$ and $Q_b$. Thus, we find the rate of change of the water surface elevation by setting the time derivative of Equation 9 equal to $Q_{in} - Q_b$ and rearranging, which yields

$$\frac{dh_w}{dt} = \frac{h_{w,0}^p}{p h_w^{p-1} V_{w,0}} (Q_{in} - Q_b).$$
(10)

We consider two scenarios for evolving the water level. In the first scenario ("flotation scenario") we assume that the effective pressure is initially zero at the basin outlet and that the basin begins to drain shortly after starting each simulation. In this scenario we set $Q_{in} = 0 \text{ m}^3 \text{ s}^{-1}$ since the basin is already full and the flood occurs soon after the simulation begins. The initial water level is

$$h_{w,0} = \frac{\rho_i}{\rho_w} (H_b - h_i)$$
(11)

and the volume of floating ice is related to its thickness by integrating Equation 6 and substituting in Equation 11:

$$V_i = \int_{h_{w,0}}^{h_i + h_{w,0}} a z_b^{p-1} \, dz_b = \frac{a}{p} \left[ \left( h_i + \frac{\rho_i}{\rho_w} (H_b - h_i) \right)^p - \left( \frac{\rho_i}{\rho_w} (H_b - h_i) \right)^p \right].$$
(12)

Since the ice volume is known (Eq. 7), $h_i$ (and therefore $h_{w,0}$) can be determined by adjusting its value until Equations 7 and

12 are in agreement. In the second scenario ("filling scenario") we set the initial water level to $h_{w,0} = 10$ m and the discharge into the lake to $Q_{in} = 20$ m$^3$ s$^{-1}$, which allows the basin to fill while draining. In both scenarios, we assume the filling rate $Q_{in}$ remains constant despite the changing climate and year-to-year variability. We tested values of 0–25 m$^3$ s$^{-1}$ and while different values of $Q_{in}$ impact the flood magnitudes and how quickly a flood is initiated, we found that varying $Q_{in}$ does not qualitatively affect our results and so we chose to keep $Q_{in}$ constant throughout the filling scenario simulations. Additionally,

$Q_{in}$ has little impact on the outburst flood hydrographs once a flood initiates because the flood discharge exceeds $Q_{in}$ by more than two orders of magnitude. Note that we only apply the filling scenario to box-shaped basins in order to avoid geometric complexities associated with raising and lowering a fragmented layer of remnant ice along a sloping basin, and therefore we compute the floating ice thickness by simply dividing the ice volume by the basin surface area. We define basin storage capacity, $V_s$, as the water volume when the basin level is at flotation depth and the peak water volume as the volume of water when the

lake in a simulation has reached peak water depth. In the flotation scenario, basin storage capacity and peak water volume are equal, however in the filling scenario, the peak water volume is less than the basin storage capacity for all simulations.

### 2.1.3   Numerics

The outburst flood model is nondimensionalized and solved numerically using methods described by Kingslake (2013) and Kingslake and Ng (2013a). We use a constant time step in dimensionless units, resulting in the dimensional time step decreasing

from $\sim$400 s to $\sim$300 s as the glacier thins and retreats. For the grid spacing we set $ds = s/100$, which equals $\sim 50$ m at year 0 and decreases as the glacier retreats. At each time step, given $S$, $h_w$, and $h_i$, we solve Equation 2 and 3 simultaneously for $N$ and $Q$, with $m$ defined by Equation 4, $dS/dt$ in Equation 2 provided by Equation 1, and the boundary condition on $N$ at the lake provided by Equation 5. Employing an approach referred to as the relaxation method by Kingslake (2013), a fictitious time-derivative is introduced to the left of Equation 2 and, after making an initial guess at the discharge at the lake $Q_0$, the result

is solved with Equation 3 using forward-time upwind difference until the fictitious derivative disappears. This is performed repeatedly within a root-finding algorithm, which tunes $Q_0$, until the $N$ boundary condition (Equation 5) is met. This results in profiles of $N$ and $Q$ which obey Equation 2 and 3 and the boundary conditions. These are used to evolve $h_w$ and $S$ forward in time with Equation 10 and Equation 1, respectively. Initial values for $h_w$ and $S$ are discussed in Section 2.3.

### 2.2   Glacier evolution

We model changes in glacier geometry with a one-dimensional form of the shallow shelf approximation (SSA), which is a depth- and width-integrated flow model (Nick et al., 2009; Enderlin et al., 2013; Carnahan et al., 2019). For our simulations, we use a glacier with a simple bed geometry (a uniformly sloping bed with a slope of 4°) and assume a simple climate parameterization. After running the model to steady-state, we invoke glacier retreat by applying a constant rate of warming. The simulations are run until the glacier terminus retreats past the basin, which is initially located 75% of the way from the

head of the glacier to its terminus. We ran additional simulations with different parameter values for bed slope, climate, and

basin location. Although these parameters affect the details of how outburst floods change from year to year, they do not affect the overall pattern of how floods evolve.

The glacier flow model is based on conservation of momentum, which requires that the glaciological driving stress is balanced by gradients in longitudinal stress, lateral drag, and basal drag (van der Veen, 2013), such that

$$2\frac{\partial}{\partial x}\left(H\nu\frac{\partial U}{\partial x}\right) - \frac{H}{W}\left(\frac{5U}{2AW}\right)^{1/3} - \tau = \rho_i gH\frac{\partial h_s}{\partial x}, \tag{13}$$

where $\nu$ is the depth- and width-averaged viscosity, $U$ is the depth- and width-averaged velocity, $W$ is glacier width, $\tau$ is the basal shear stress, and $h_s$ is the glacier surface elevation. The viscosity depends on the strain rate according to Glen's Flow Law:

$$\nu = A^{-1/3}\left|\frac{\partial U}{\partial x}\right|^{2/3}. \tag{14}$$

We assume a simplified ad hoc parameterization of the basal shear stress, in which $\tau = \tau_{max}(U/\max(U))$ with $\tau_{max} = 2.5 \times 10^5$ Pa. This parameterization results in shear stresses on the order of $10^5$ Pa, which are typical values for valley glaciers (e.g., Brædstrup et al., 2016), and produces realistic glacier geometries and velocities across a wide range of bed slopes and climates. Importantly, the parameterization ensures that the resistive stresses never exceed the glaciological driving stress. For boundary conditions, we prescribe a velocity of $U = 0$ at the ice divide ($x = 0$) and velocity gradient $\partial U/\partial x = 0$ at the
terminus ($x = L$).

The glacier surface is updated using a depth- and width-integrated mass continuity equation (van der Veen, 2013), in which

$$\frac{\partial H}{\partial t} = \dot{B} - \frac{1}{W}\frac{\partial(UHW)}{\partial x}, \tag{15}$$

where $\dot{B}$ is the width-averaged specific mass balance rate. We prescribe the mass balance rate by using a constant mass balance gradient and imposing a maximum balance rate $\dot{B}_{max}$ (as is commonly observed; e.g., Van Beusekom et al., 2010). In other
words,

$$\dot{B}(z) = \min\left(\frac{d\dot{B}}{dz}(z - \mathrm{ELA}), \dot{B}_{max}\right), \tag{16}$$

where ELA is the equilibrium line altitude. We use an initial ELA of 1500 m, a balance gradient of $d\dot{B}/dz = 0.005$ a$^{-1}$, and a maximum balance rate of $\dot{B}_{max} = 2$ m a$^{-1}$. The ELA increases at a rate of 5 m a$^{-1}$ to approximate expected changes under climate warming scenarios (Huss and Hock, 2015).

The model equations are discretized following the methodology described in Enderlin et al. (2013) using finite differences for the spatial discretization (initial grid spacing of 100 m), a staggered, moving grid, and a forward Euler time step ($\Delta t = 0.05$ a). At each time step, Equation 13 is solved for the velocity and the glacier surface is adjusted according to Equation 15. The glacier length is updated by allowing the terminus to advance at its flow speed, and any ice thinner than 0.1 m is subsequently removed from the domain.

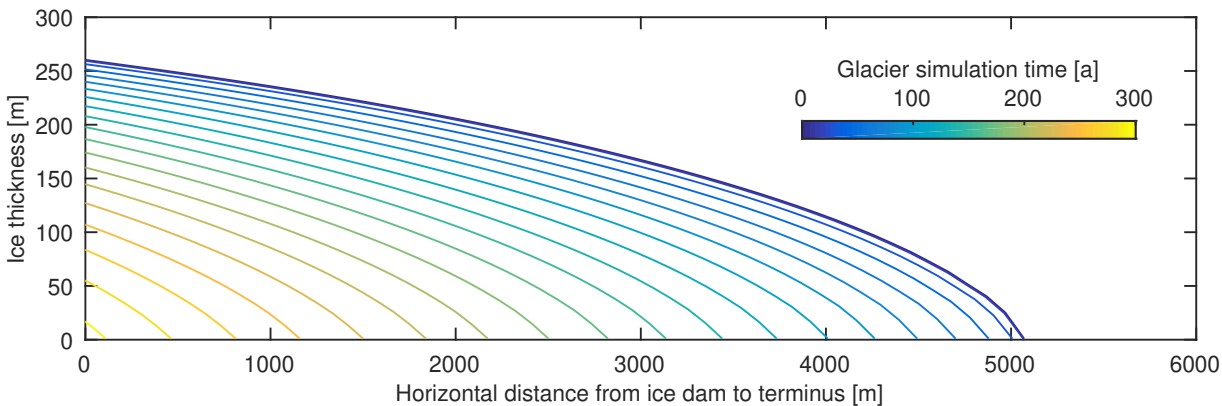

**Figure 3.** Ice thickness profile from the marginal basin to the terminus over 300 years of glacier retreat.

## 2.3  Simulations

In the glacier flow model it takes about 300 years for the glacier terminus to retreat from its initial position to the location of the marginal basin, a distance of $\sim$5 km (Fig. 3). For each year of the glacier model output, we extract (i) the distance from the basin to the terminus, which we take to equal the conduit length, (ii) the glacier thickness profile and ice dam thickness, and (iii) the specific mass balance rate of the ice dam. Then (i) and (ii) are fed directly into the outburst flood model and (iii) is used to calculate the volume of floating ice remaining in the basin (Fig. 4).

To demonstrate how remnant ice affects outburst floods, we first run simulations in which we use the glacier geometry from one time step in the glacier flow model, assume a box-shaped basin, and run the outburst flood model with different starting water volumes. We run the simulations both without ice and with enough ice to force ice dam flotation. Thus these initial simulations are similar to those that we run in the flotation scenario (next paragraph) except that here the basin is not necessarily at flotation depth unless it contains remnant ice.

We then use the evolving glacier and basin geometries to model long-period variations in outburst floods using the flotation and filling scenarios described in Section 2.1.2. In the flotation scenario, we assume that the initial water pressure at the basin outlet equals the overburden pressure of the ice dam. In this scenario, the initial conduit area is 1 m$^2$. Thus we assume that the basin is not connected to the subglacial drainage system until the onset of the outburst flood. To test the effect of basin geometry and floating ice on outburst flood evolution, we run simulations with (i) box-shaped, wedge-shaped, and semicircular-cone-shaped basins and (ii) both with and without floating ice (Fig. 5). For the box-shaped basin we used a value of $a = 8.5 \times 10^5$ m$^2$, for the wedge-shaped basin we used a basin width of 1910 m and a bed slope of 15°, and for the semicircular-cone-shaped basin we used a bed slope of 10.6°. These values were chosen so that the basins would initially have the same basin storage capacity $V_s$.

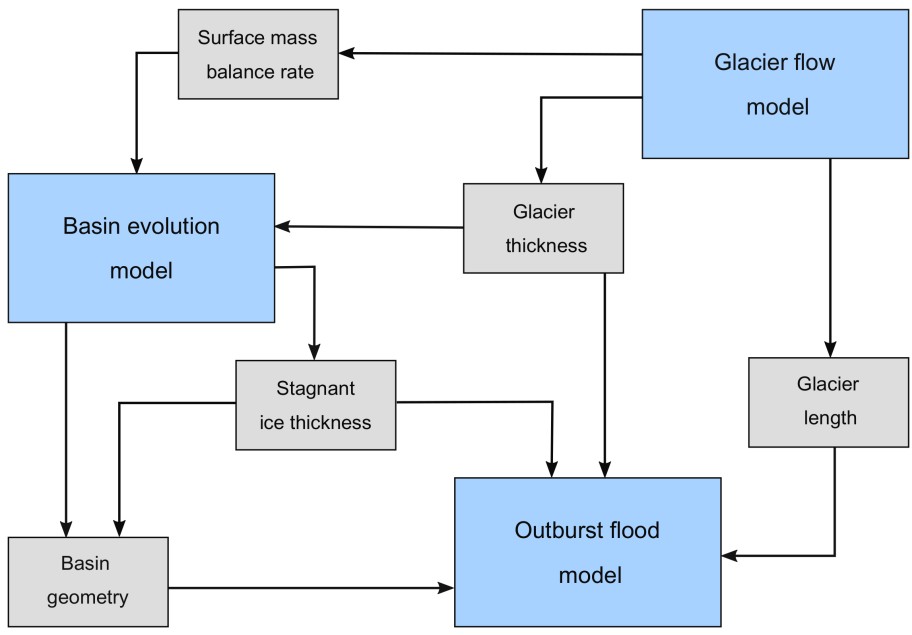

**Figure 4.** Information flow between the glacier flow, basin evolution, and outburst flood models.

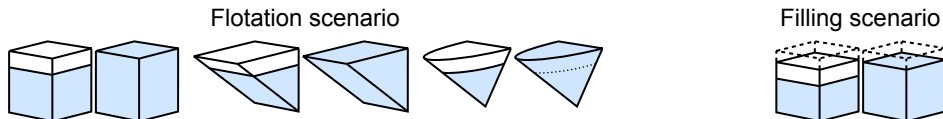

**Figure 5.** Schematic of the various scenarios that we considered in our simulations, illustrating basin geometry, presence/absence of remnant ice, and whether drainage initiated when the lake reached flotation depth or while the basin was filling. For the flotation scenario, we prescribe $Q_{in} = 0 \text{ m}^3\text{s}^{-1}$, $S_0 = 1 \text{ m}^2$, and $h_{w,0} = (H_b - h_i)\rho_i/\rho_w$ (flotation depth). For the filling scenario simulations, we prescribe: the discharge into the basin as $Q_{in} = 20 \text{ m}^3\text{s}^{-1}$, the initial conduit cross-sectional area as $S_0 = 0.1 \text{ m}^2$, and the initial water level $h_{w,0} = 10 \text{ m}$. The basins all have the same initial volume (year 0 in the ice flow model), but as the ice dam thins the changes in basin volume are nonlinear for the cone- and wedge-shaped basins. The simulations with remnant ice assume the basin is completely filled with ice at year 0.

In the filling scenario we prescribe a small initial water level of 10 m and an initial conduit area of 0.1 m$^2$. The subglacial conduit is connected to the marginal basin as filling occurs and the conduit therefore evolves prior to the onset of the outburst flood, which occurs naturally once $Q_b > Q_{in}$. The granular nature of the floating ice makes a full treatment of its behavior during filling and drainage nontrivial. The floating ice should gradually expand outward as the basin fills, but then friction should prevent it from flowing back down to the bottom of the basin during rapid drainage. In the filling scenario the basin

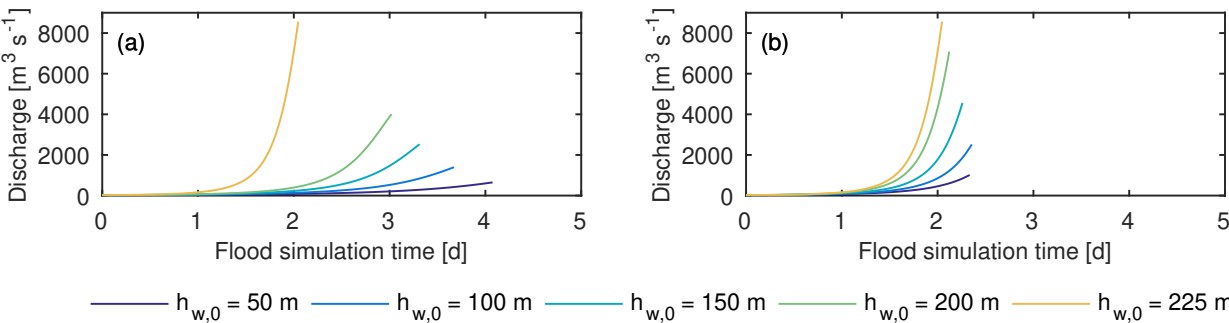

**Figure 6.** Demonstration of the impact of floating ice on outburst flood hydrographs for an ice dam height of 250 m (year 50 in the glacier simulations). The glacier geometry and basin shape (box) are the same in all simulations. In (a), the initial water height, $h_{w,0}$, is varied and there is no floating ice in the basin. The initial water heights are the same in (b) except that enough floating ice is added to force ice dam flotation. For $h_{w,0} = 225$ m, the water level is equal to flotation depth so there is no remnant ice and the curves in (a) and (b) are the same. Note that the modeled hydrographs do not include the rapidly falling limb of the floods because the outburst flood model is not capable of handling open channel flow, which occurs when the basin water level drops below the conduit roof.

generally does not fill up completely, greatly complicating the task of tracking the thickness and location of the floating ice except when the basin walls are vertical. For this reason we only apply the filling scenario to box-shaped basins.

## 3 Results

For glaciers with a fixed geometry, floating ice in a basin causes outburst floods to have higher peak discharge and shorter duration than might otherwise be expected based solely on the consideration of flood water volume (Fig. 6). Consequently, changes in remnant ice volume impact the evolution of glacier outburst floods over decadal to centennial timescales. In our transient glacier simulations we observed similar trends in flood hydrographs regardless of basin hypsometry and whether the simulations started with the basins filled to flotation depth (Fig. 7) or if the basins were connected to the subglacial hydrological system as they filled (Fig. 8). The floods that occur in the years immediately following the formation of a marginal basin have low peak discharge on account of the basin's small storage capacity. As the climate warms, the remnant ice thins more quickly than the ice dam, which is partially replenished by the delivery of ice from upstream. The largest outburst floods occur when the basin becomes ice free, after which the peak discharge decreases until the basin is no longer dammed by the trunk glacier.

For the simulations in which the basins were filled to flotation depth before flood onset, we considered three different basin hypsometries (semicircular-cone-, wedge-, and box-shaped) that had identical storage capacities at the time of basin formation (year 0). The cone-shaped basin produced the largest outburst floods in terms of peak discharge, however flood magnitude decreased more rapidly across the simulations for the cone-shaped basin compared to the wedge- and box-shaped basins (Fig.

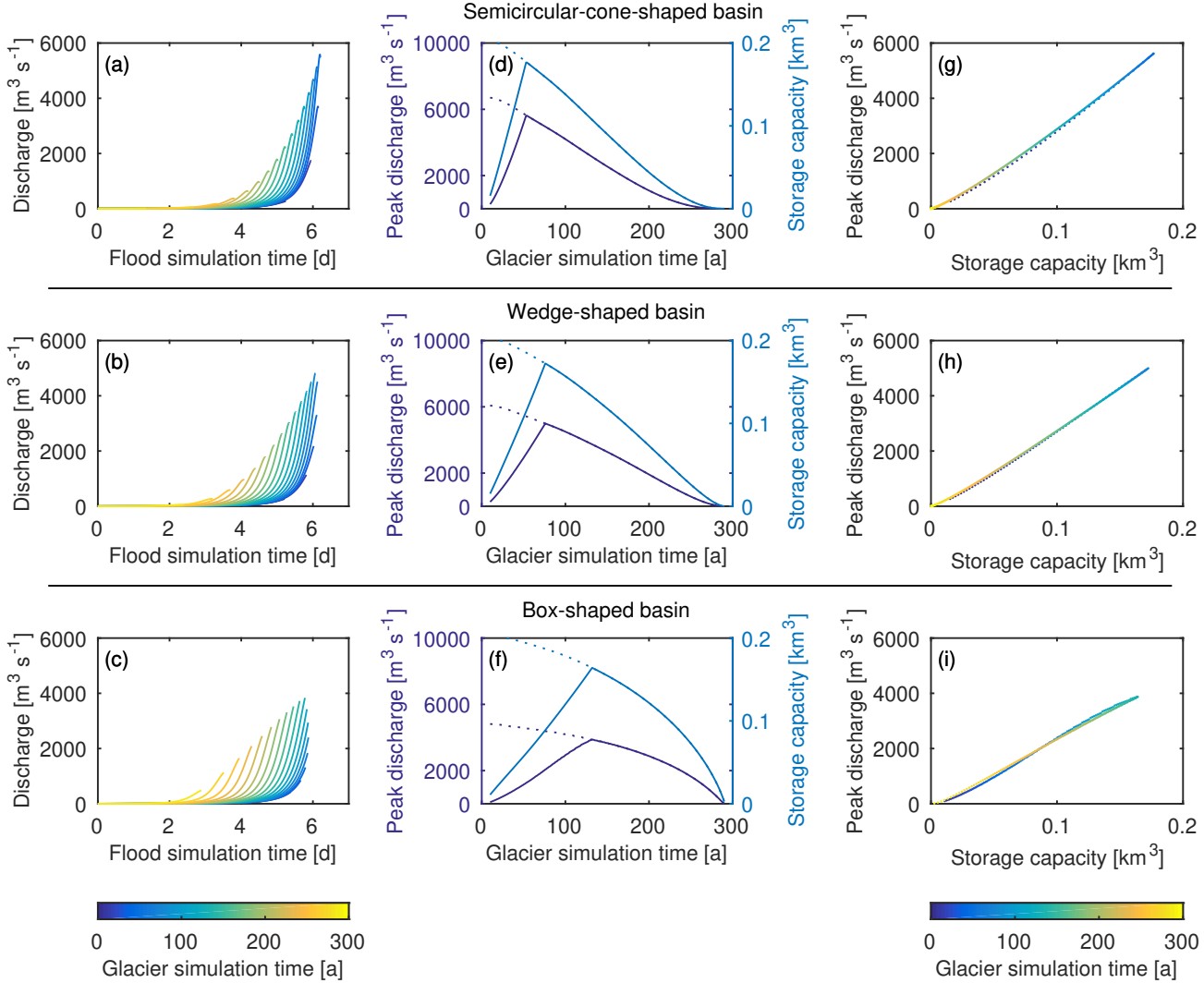

**Figure 7.** Comparison of annual outburst floods for semicircular-cone-, wedge-, and box-shaped basins for the simulations in which the basin is initially at flotation depth. Left panels (a,b,c): annual outburst flood hydrographs when the basin is initially filled with ice. Middle panels (d,e,f): peak discharge and storage capacity over time. The fork in the early years of the simulations represents ice-filled (solid line) and ice-free (dotted line) scenarios. Right panels (g,h,i): Peak discharge vs. storage capacity for the ice-filled scenario. We refer to the timescale of the glacier flow model as "glacier simulation time" and the timescale of the outburst flood model as "flood simulation time".

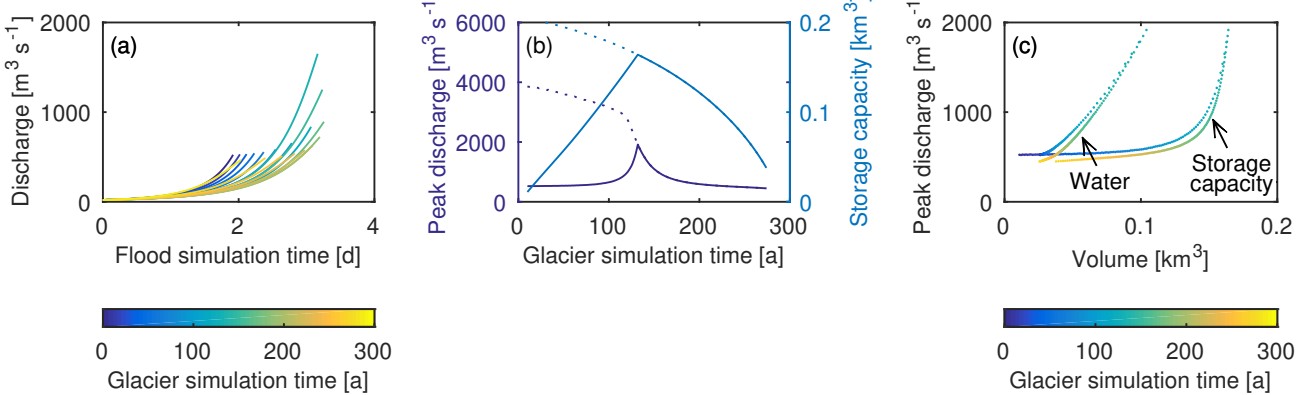

**Figure 8.** Comparison of annual outburst floods for the box-shaped basin in which the basin is connected to the subglacial hydrological system as it fills. (a) Annual outburst flood hydrographs. (b) Peak discharge and basin storage capacity for ice-filled and ice-free basins. The fork in the early years of the simulations represents ice-filled (solid line) and ice-free (dotted line) scenarios. (c) Relationship between peak discharge and peak water volume ("Water") and basin storage capacity ("Storage capacity").

7a–c). The differences in peak discharge and duration of large magnitude floods arise because, owing to their hypsometry, cone-shaped basins lose their floating ice more rapidly than wedge- or box-shaped basins and because as they drain the floating ice in the basin exerts pressure on the underlying water that helps to drive the water out of the basin. However, ice-dam thinning reduces basin capacity faster in cone-shaped basins than in wedge- or box-shaped basins and therefore the magnitude of the outburst floods in cone-shaped basins decrease more rapidly. In the early years of the simulations, flood magnitude increases in basins that are initially filled with ice (solid line) until the basin is ice free, whereas in basins that are initially ice-free (dotted line) the flood magnitude always decreases (Fig. 7d–f). For all three hypsometries we observe a nearly linear relationship between peak discharge and storage capacity (Fig. 7g–i). This relationship holds regardless of whether the basin contains ice or is ice-free.

For the simulations in which the basin is initially drained of water but remains connected to the subglacial hydrological system as the basin fills, the relationship between peak discharge and peak water volume (which is often less than the storage capacity, as defined above) takes a slightly different form. First, the basin often does not reach flotation depth in our simulations because the conduit enlarges at the same time as the basin is filling and consequently the outburst floods tend to be smaller in magnitude (Fig. 8). This behavior is sensitive to the model parameters though, as the basin could be made to reach or even exceed flotation depth by selecting a larger influx $Q_{in}$. Second, there is a more prominent spike in the peak discharge curve that occurs as the remnant ice is about to melt away completely (Fig. 8b). Similar to the flotation scenario, the relationship between peak discharge and peak water volume is approximately linear; however, in the filling scenario the peak water volume is less than the storage capacity because the basin does not completely fill (Fig. 8c). As a result the relationship between peak discharge and (total) storage capacity is not linear.

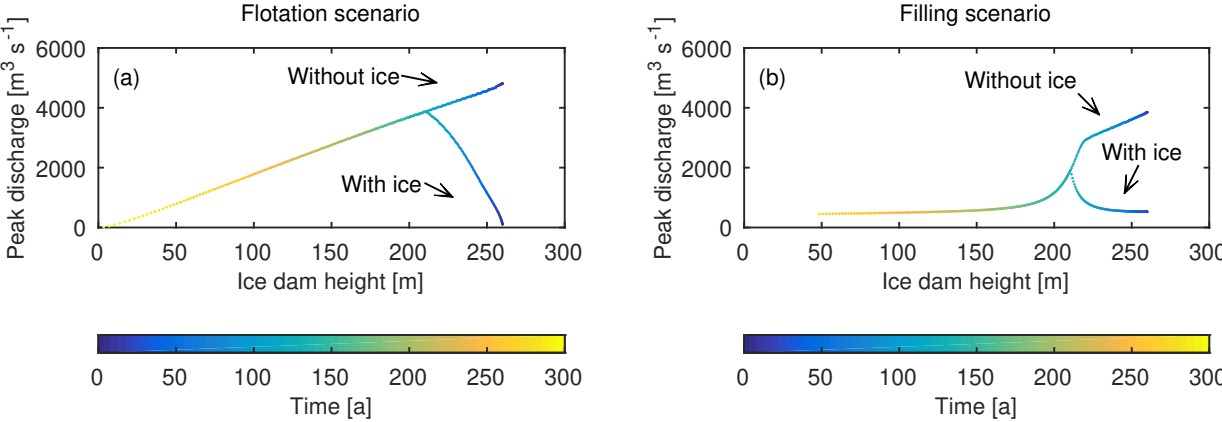

**Figure 9.** Comparison of peak discharge and ice dam height for the box-shaped basin under the (a) flotation scenario and (b) filling scenarios. "With ice" indicates the relationship between peak discharge and ice dam height when remnant ice is accounted for in the model, whereas "Without ice" indicates the relationship when remnant ice is neglected.

Figures 7 and 8 show that remnant ice can act to produce similar size outburst floods for very different glacier thicknesses. To further illustrate this consequence of remnant ice, we plot peak discharge versus ice dam height for the box-shaped basin in both the filling and flotation scenarios (Fig. 9). In the flotation scenario we observe large variability in outburst floods during glacier recession; for example, a peak discharge of $2000 \ \mathrm{m^3 \ s^{-1}}$ occurs when the ice dam height is 240 m and then again when it is 120 m. In contrast, in the filling scenario, the peak discharge is nearly independent of ice dam height except during the years in which the basin becomes ice-free (when the ice dam height was around 210 m). On the other hand, when remnant ice is excluded from the simulations, the peak discharge increases monotonically with ice dam height in both the flotation and filling scenarios. Thus, proper accounting of remnant ice is critical for quantifying the evolution of outburst floods over decadal to centennial timescales.

## 4 Discussion

### 4.1 Impact of remnant ice and ice flow on basin storage capacity

During decadal to centennial scale glacier retreat, the peak discharge of ice-dammed outburst floods will tend to increase with time as long as there is remnant ice in a basin that is melting away. The peak discharge will begin to decrease only once the remnant ice is gone. This result is independent of basin geometry and the mechanism of drainage onset and is ultimately a consequence of the proportionality between peak discharge and basin storage capacity that occurs for individual basins despite large changes in glacier geometry and remnant ice. In other words, the model exhibits very little hysteresis between peak discharge and basin storage capacity (Figs. 7g–i and 8c). As a result, the time rate change of the basin storage capacity

illuminates how peak discharge evolves with time. The storage capacity is found by inserting Equation 11 into Equation 8, which gives

$$V_s = \frac{a}{p} \left( \frac{\rho_i}{\rho_w} (H_b - h_i) \right)^p .$$
(17)

Taking the derivative of Equation 17 with respect to time, we find that storage capacity evolves according to

$$\frac{dV_s}{dt} = \left[ a \left( \frac{\rho_i}{\rho_w} (H_b - h_i) \right)^{p-1} \right] \left( \frac{dH_b}{dt} - \frac{dh_i}{dt} \right) .$$
(18)

The term in square brackets in Equation 18 is always positive, and thus the storage capacity will always increase as long as $dH_b/dt > dh_i/dt$ (i.e., the ice dam is thinning less quickly than the remnant ice). Thinning of the ice dam due to surface melting is partially offset by ice flow from upglacier and therefore the storage capacity, and by extension the peak discharge of 285 outburst floods, will continue to increase until a basin is ice free.

However, in our simulations we did not account for ice flow or calving of icebergs into a basin, which would require a significantly more sophisticated ice flow model. Ice flow into a basin shortens the basin and reduces the storage capacity. Calving changes the basin geometry but tends to have little net impact on storage capacity because it has two competing effects: it results in retreat of an ice dam away from a basin, which increases storage capacity, but it also adds to the volume of 290 remnant ice, which reduces storage capacity. For a more detailed discussion on the impacts of ice flow and calving on storage capacity, see Appendix A.

Our analysis here has focused solely on basin storage capacity. The relationship between peak discharge and water volume is likely to become more complicated than presented in Figures 7 and 8 when changes in basin geometry due to ice flow and calving are accounted for. Moreover, we do not account for lateral variations in glacier thickness that may cause the seal of the 295 ice dam to be located some distance from the basin. Flow re-direction toward a marginal basin due to lateral surface gradients will affect the ice dam thinning rates and location of the seal in ways that we are unable to capture in our one-dimensional flowline model. These additional complexities should be considered in more detail in future studies.

## 4.2 Comparison to the Clague-Mathews relationship

Observations across a range of systems are suggestive of a power-law relationship between the peak discharge and total water 300 volume drained, $\Delta V_w$, during outburst floods (Clague and Mathews, 1973; Walder and Costa, 1996):

$$Q_{peak} \propto \Delta V_w^{2/3}.$$
(19)

This relationship is commonly referred to as the Clague-Mathews relationship. Ng and Björnsson (2003) examined the Clague-Mathews relationship by analyzing the equations describing flood evolution. Using a simplified version of the outburst flood model used in this study, they demonstrated that for individual basins that do not drain completely, (i) each flood trajectory has 305 a unique set of initial and final water levels and peak discharge, (ii) peak discharge monotonically increases with water volume, and (iii) there is a power-law relationship between discharge and water volume for floods. They focused on analyzing basins

that experience incomplete drainage because some information on flood mechanics is lost if a basin drains completely. Their analysis predicts an exponent in the power-law relationship of about 1–2 for individual basins, depending on basin geometry and ice coverage. When observed flood data from multiple glaciers was scaled and placed into their theoretical framework, they arrived at an exponent close to 1. They hypothesized that the difference between their theoretical exponent and the exponent in the Clague-Mathews relationship is due to confounding factors such as differences in flood initiation, basin geometry, and complete drainage.

Our simulations extend the work of Ng and Björnsson (2003). We modeled variations in outburst floods over decadal to centennial timescales, from different shaped basins, and with different drainage scenarios (flotation vs. filling). In addition, in our simulations the basins always drained completely. We observe that the relationship between peak discharge and peak water volume reached (equal to volume drained) is nearly linear in the flotation scenario (power law exponent of ∼1; Fig. 7g–i) and superlinear in the filling scenario (power law exponent >1; Fig. 8c). These trends occur regardless of whether the basins contain remnant ice, in which case peak discharge and storage capacity increase with time, or are ice free. We also find that the slopes of the discharge-volume curves depend on basin geometry, where basins that contain less volume near their outlets produce a steeper slope. This is likely a result of cone- and wedge-shaped basins being able to maintain high water pressures as they drain, thus favoring more rapid conduit growth.

The explanation for the lower, 2/3 exponent in the Clague-Mathews relationship remains elusive. Ng and Björnsson (2003) suggest that the lower exponent is due to differences in flood initiation across different basins (implying that flood initiation may depend on basin hypsometry). Flood initiation could also depend on some time-varying property such as ice dam thickness or the size of a previous year's flood, both of which could influence the state of the subglacial hydrological system at the onset of a flood (see also Kingslake, 2015). For example, in some flood events involving large volumes of water (i.e., when the basin storage capacity is large) have a persistent impact on the subglacial system, so that when a basin refills it does so while slowly draining, whereas floods involving small volumes of water may have a less persistent impact and as a result subsequent floods will only initiate after the basin reaches flotation depth.

## 4.3  Hazard assessment confounded by poor understanding of drainage onset

Mitigating risks due to outburst floods requires accurate predictions of flood initiation, peak discharge, and flood duration. As our results show, these properties depend on basin hypsometry and the amount of remnant ice in a basin, which may be unknown in many situations, making it difficult to assess current and future outburst flood hazards. In contrast, changes in ice dam thickness are much easier to observe making it is tempting to try to relate ice dam thickness to potential flood magnitudes. However, our simulations (Figs. 7 and 8) suggest that similar size outburst floods may occur for very different ice dam thicknesses if a basin contains remnant, floating ice. This nonlinearity occurs both for basins that do not connect to the hydrological system until drainage onset (flotation scenario) and for basins that remain connected to the subglacial hydrological system during filling (filling scenario) (Fig. 9).

Remnant, floating ice affects outburst floods in multiple ways that also affect hazard assessment. First, the presence of floating ice reduces the storage capacity of a basin (Figs. 7d–f and 8b). Shortly after a basin forms, the presence of remnant ice

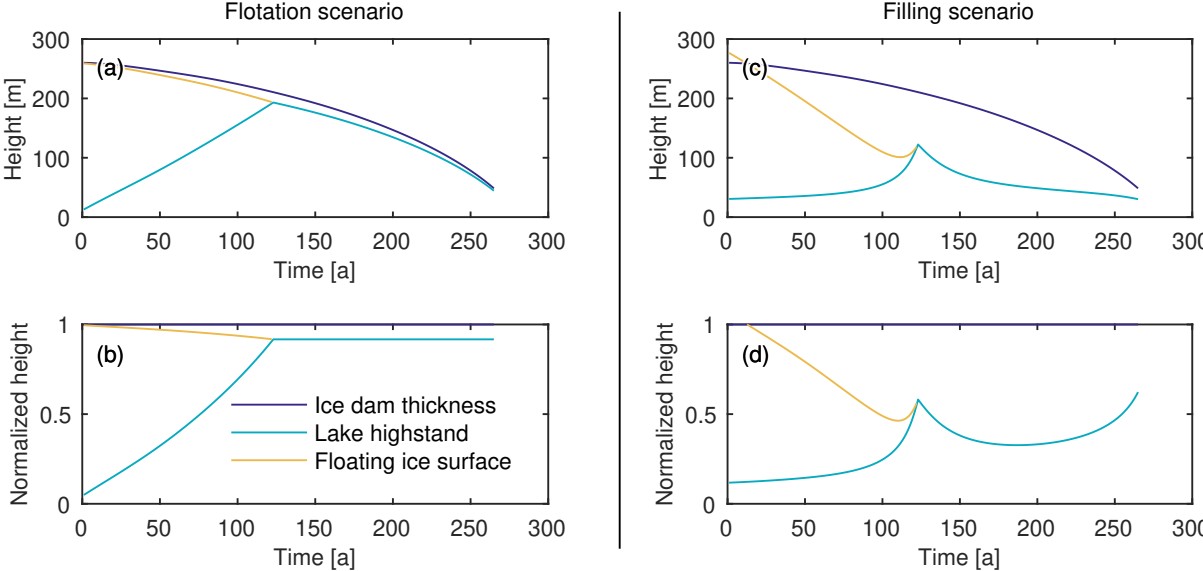

**Figure 10.** Time series of the ice dam elevation, lake highstand (peak water surface), and floating ice surface elevation at lake highstand for the (a) flotation and (b) filling scenarios. In (c) and (d), the values for height are normalized using the ice dam thickness as a measure of scale.

limits the storage capacity and causes the peak discharge to be small. As the ice melts over time the storage capacity and peak discharge increase until the basin is ice-free. The relationship is more clearly seen in the flotation scenario than in the filling scenario (Fig. 10). Floods tend to be more uniform from year to year in the filling scenario because when floating ice is present, the additional pressure causes the outlet conduit to open relatively quickly and the basin drains before filling to flotation depth.
Further, once the floating ice has melted and the ice dam is thinner, the ice-overburden pressure is less and the creep closure is slowed down by the increasing water pressure as the basin is filling. As a result, melt opening overcomes creep closure which acts to initiate a flood more quickly, and again the basin only fills partially (Fig. 10c–d).

    A second consequence of floating ice is that it affects the duration of outburst floods (Fig. 11). The role of floating ice is again most clear in the flotation scenario. Early in the simulations, when the storage capacity is small, outburst floods can occur
that have higher peak discharge than might be expected because the pressure from the floating ice helps to drive water out of the basin. However, the small amounts of water (relative to the size of the basin) in these events are not able to melt the conduit walls as rapidly as later floods and the overburden pressure from the ice dam, which favors creep closure, is high. Consequently the floods tend to be slower building and the basins may take about a week to drain. Later, when the floating ice is gone and the ice dam is also thinner, drainage can proceed more quickly. This creates challenges for flood risk mitigation because floods
with similar peak discharges may occur over timescales of a few days (e.g., Anderson et al., 2003) to a week (e.g., Huss et al., 2007) depending on basin conditions (Fig. 11a). For basins that fill while connected to the subglacial hydrological system, there tends to be less variability in the duration of outburst floods (Fig. 11b).

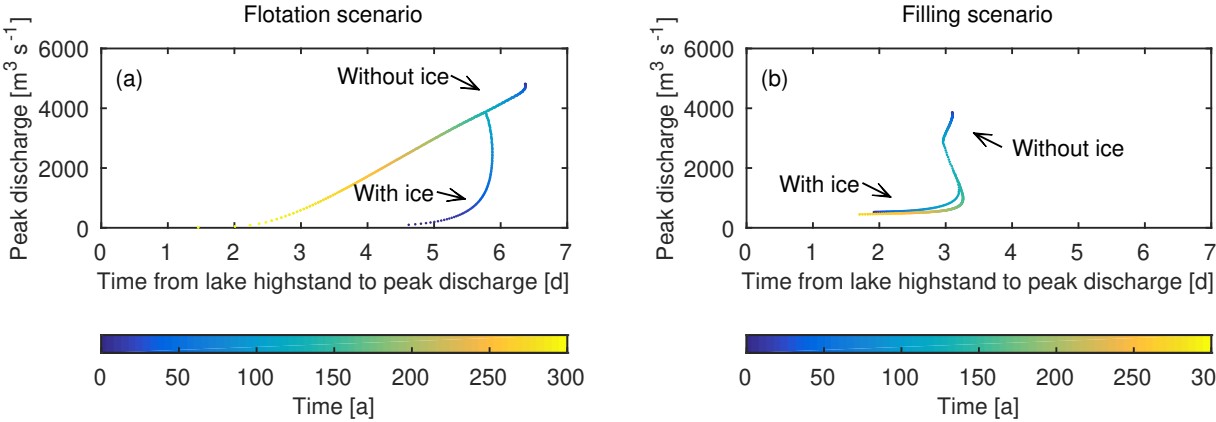

**Figure 11.** Comparison of peak discharge and time from lake highstand to peak discharge for the box-shaped basin under the (a) flotation and (b) filling scenarios.

Our simulations predict differences in how outburst floods will evolve with time, depending on whether a basin begins to drain once it has filled or if the basin remains connected to the subglacial hydrological system and begins to drain as it is filling. Furthermore, our model does not address the large year-to-year variability in peak discharge and total water volume of outburst floods, which may vary by a factor of two or more in subsequent years (e.g., Huss et al., 2007; Neal, 2007; Kienholz et al., 2020) and is also likely related to the onset mechanism. A deeper understanding of the onset of outburst floods is therefore critical to improving our ability to assess both the short- and long-term risk associated with outburst floods.

## 5 Conclusions

We modeled the effect of changes in glacier and basin geometries on the magnitude and duration of ice-dammed glacier outburst floods. In our simulations we accounted for remnant, floating ice that is left behind in marginal basins during the retreat of tributary glaciers. The remnant ice acts to exerts pressure on the underlying water and thus helps to increase discharge by enlarging the subglacial conduit. Because the remnant ice is not replenished by ice flow from upglacier, it thins more quickly than the adjacent ice dam. As a result the basin storage capacity increases with time as long as the basin contains remnant ice, regardless of basin hypsometry. Despite complex relationships between glacier and basin hypsometry, remnant ice thickness, and discharge, we find nearly linear relationships between peak outburst flood discharge and total water volume for individual basins. This is regardless of whether the basin drains once it reaches flotation depth or if it remains connected to the subglacial hydrological system while filling. However, differences in modeled outburst floods for the two different drainage scenarios that we considered highlight the importance of improving our understanding of drainage onset. Basins that are continuously connected to the subglacial drainage system tend to produce similar outburst floods from one year to the next, except during

the years immediately before and after the loss of remnant ice, whereas basins that do not begin to drain until full produce much larger variability in the magnitude and timing of outburst floods.

In our simulations we made a number of simplifying assumptions in order to garner a fundamental understanding of the long-period variability of outburst floods in an evolving catchment. In particular, we (1) assumed that the seal of the ice dam was immediately adjacent to the basin and did not account for changes in the hydraulic potential gradient that could drive water from the glacier into the basin as it is filling, (2) treated remnant ice as fluid that spreads out as a basin fills, instead of accounting for the granular nature of the icebergs, (3) did not consider the state of the glacier's hydrological system at the time of drainage, which may impact flood evolution, or changes in ice flow due to the evolving subglacial hydrology, (4) did not allow for ice flow into the basin from the trunk glacier, and (5) did not account for interannual variability in climate and its affects on glacier geometry and basin filling rates. Year-to-year variability in the timing, duration, and magnitude of outburst floods (e.g., Huss et al., 2007; Neal, 2007; Kienholz et al., 2020) may mask the longer period changes in outburst floods due to changes in glacier and basin geometry that we modeled here. Additional and more sophisticated modeling studies will be needed to elucidate the impact of these processes on the decadal and centennial evolution of outburst floods and to connect outburst floods to landscape and ecosystem evolution.

## Appendix A: Effect of ice flow and calving on basin storage capacity of a box-shaped basin

For a box-shaped basin, the basin storage capacity is given by,

$$V_s = \frac{\rho_i}{\rho_w}(H_b - h_i)W_b L_b. \tag{A1}$$

Since we are now allowing for ice flow and calving, the basin length is no longer treated as a constant and the remnant ice thickness varies in response to the addition of new icebergs and compaction/extension due to changes in the location of the ice dam. The rate of change of the storage capacity is

$$\frac{dV_s}{dt} = \frac{\rho_i}{\rho_w}\left(\frac{dH_b}{dt} - \frac{dh_i}{dt}\right)W_b L_b + \frac{\rho_i}{\rho_w}(H_b - h_i)W_b\frac{dL_b}{dt}. \tag{A2}$$

The thickness of the remnant ice changes at a rate that is given by

$$\frac{dh_i}{dt} = \dot{B}_b + U_c\frac{H_b}{L_b} - \frac{h_i}{L_b}\frac{dL_b}{dt}, \tag{A3}$$

where $U_c$ is the calving rate. The three terms on the right-hand side of Equation A3 describe the changes in ice thickness due to the surface mass balance, the influx of freshly calved ice, and changes in the ice dam location. The rate of change of the basin length is simply $dL_b/dt = U_c - U_b$, where $U_b$ is the rate at which ice is flowing toward the basin. By inserting these expressions for $dh_i/dt$ and $dL_b/dt$ into Equation A2 and rearranging, we find that

$$\frac{1}{W_b}\frac{\rho_w}{\rho_i}\frac{dV_s}{dt} = \frac{dH_b}{dt}L_b - \dot{B}_b L_b - U_b H_b, \tag{A4}$$

which indicates that the storage capacity will increase as long as $dH_b/dt > \dot{B}_b + U_b H_b/L_b$. For a box-shaped basin the effects of calving cancel out completely and changes in storage capacity are only due to thinning of the ice dam, the surface mass

balance rate, and the ice flux toward the basin. The effect of ice flow is to reduce the maximum storage capacity that occurs in a basin and to increase the time that it takes for the maximum storage capacity to be reached since contraction of the remnant ice reduces the surface area that is susceptible to melting.

Equation A4 illustrates that ice flow toward a basin may have important consequences for basin storage capacity. For exam-
ple, the remnant ice in Suicide Basin, the source of recent outburst floods at Mendenhall Glacier, has a surface mass balance flux ($\dot{B}_b L_b W_b$) of about $-2.5 \times 10^6$ m$^3$ a$^{-1}$ and the ice flux toward the basin ($U_b H_b W_b$) is roughly 3.5–7.0$\times 10^5$ m$^3$ a$^{-1}$ (both expressed as ice equivalent) (Kienholz et al., 2020); thus ice flow is currently offsetting the growth in storage capacity due to melting by about 25%.

*Code availability.*  MATLAB script files for full model are available at https://zenodo.org/record/5488047.YUFKutNKit9.

*Author contributions.*  JA and EH conceived the study, AJ ran the simulations with assistance from JA and JK, AJ and JA prepared the manuscript with contributions from JK and EH.

*Competing interests.*  The authors declare that they have no conflict of interest.

*Acknowledgements.*  This project was supported by funding from the Alaska Climate Adaptation Science Center and the US National Science Foundation (OIA-1757348 and OPP-1743310). We thank C. Kienholz for fruitful discussions that led to this study.

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
