# Peer review of "Long-period variability in ice-dammed glacier outburst floods due to evolving catchment geometry"

_The Cryosphere, 2021_

## Author Comment (AC1)

Thank you for your thorough and thoughtful feedback that will help us improve our manuscript.

This manuscript presents a modelling study of the evolution of jökulhlaups from an ice marginal lake as the damming glacier recedes. One novel aspect is (besides the retreating glacier) that it investigates how the floods are affected by remnant ice within the lake basin, due to a tributary glacier disconnecting from the main trunk. The study conducts synthetic experiments with a coupled jökulhlaup and ice dynamics model, both sub-models are of relatively low complexity, which is suitable for the paper's aim, namely, to investigate dynamics of the system in a qualitative fashion. (As an aside: note that fully quantitative modelling of jökulhlaups is not yet possible.)

The topic of the manuscript is interesting and timely, as many new glacier lakes are expected to form under a warming climate. It is well written and is suitable for publication in The Cryosphere once the minor revisions I outline below are incorporated.

We have responded to each of your comments below.

Specific comments

I am a bit confused about the importance of Q_in for the filling scenario. The authors write around line 131 that the choice of Q_in does not impact the results much (which is, by the way, opposite to the finding of Ng & al (2007, doi:10.1029/2007GL031426)). However, then in the Results (line 225) they state "This behavior is sensitive to the model parameters though, as the basin could be made to reach or even exceed flotation by selecting a larger influx Qin." This seems to me like a pretty big impact (and indeed the one I would expect). The authors should state prominently that setting Q_in (the initial channel size S0 has a similar effect) sets how full the lake will get and, consequently, the time and size of the outburst. If the authors feel adventurous, then I would suggest a plot like Fig.6b using a range of S0 (or Qin, although S0 is more arbitrary than Qin, thus I would use that).

You are correct. The parameter Qin does impact how quickly the lake fills and the conduit size at the flood onset, which therefore affects the timing and magnitude of the floods. We will change the wording here to make it more clear that the model depends on Qin but in a predictable way.

The authors should briefly explain the difference to the Kessler&Anderson (2004, doi:10.1029/2004GL020622)-model (which is what Schoof 2020 uses) who combine a cavity with an R-channel. In particular, that model would avoid having to set an arbitrary initial channel size, albeit, by having to set an almost as arbitrary bedrock bump height.

All in all, I think the model used here is totally adequate but still the difference to K&A (2004) should be stated.

Agreed. We will add these changes to section 2.1.

I think the work on Hidden Creek lake (Kennicott Glacier) should be cited, it is one of the best studied jökulhlaups and also from an ice-marginal lake (albeit, not from a de-glacierized tributary). The generic reference would be Anderson et al (2004, doi:10.1029/2002JF000004). But it might be worth checking whether another of their papers would be more relevant.

We will add this reference to line 21.

The authors need to acknowledge that drainage by flotation probably involves other processes than just channel enlargement, the work of Flowers & al (2004, doi:10.1029/2003GL019088) springs to mind.

Thanks for directing us towards this citation. We will acknowledge other processes such as increased basal water pressure and the evolution of the subglacial system which occur as a result of flotation and cite Flowers et al. (2004).

Discussion 4.1: here I struggled with this lengthy discussion on something which the model is not capturing, namely ice flow into the basin (lines 255-287). I agree that model limitations should be discussed but starting off the Discussion with this and in particular at this great depth I found a bit much. My suggestion is to move all the mathematics (Eq19-22) and much of the accompanying text into the Appendix and then briefly discuss the limitation in words here and refer to the Appendix for details.

Thank you for this suggestion. We agree this part of the discussion should be rearranged. In order to address this, we will move Eq 19-22 and the relevant accompanying text to the Appendix.

Line by line comments

2: "Marginal basins can form..." as there are other ways that marginal basins form.

Agreed.

Introduction: I think almost all citations need an "e.g." as they are not exhaustive.

Okay. We will add 'e.g.' to relevant citations in the introduction.

19: also cite Werder & al (2010, www.nat-hazards-earth-syst-sci.net/10/227/2010/) here (add the "e.g.")

We will add this reference as suggested.

18: there is a new study by Mölg & al (2021, doi:10.1002/esp.5193) for the Swiss Alps which should be cited here.

Agreed. Thank you for pointing us to this reference.

21: how do proglacial temperatures change rapidly during a flood? Flood water is as cold as usual proglacial water.

Outburst floods have been observed to cause rapid drops in water temperature in both proglacial rivers and fjords. Figure 11 in Neal (2007) shows water temperatures in the Taku River dropping by about 3 deg C during outburst floods, and Meerhoff et al. (2019) report even larger temperature drops in the Baker River in Patagonia. Kjeldsen et al. (2014) report similar observations for a fjord system in Greenland. We will expand this sentence and add the Neal and Kjeldsen references.

Neal (2007): https://pubs.usgs.gov/sir/2007/5027/pdf/sir20075027.pdf

Kjeldsen et al. (2014): https://agupubs.onlinelibrary.wiley.com/doi/10.1002/2013JF003034

26: I'm pretty sure this study was "caused" by the Kienholz & al. (2020) study. Why not acknowledge that with an extra sentence or two here?

Agreed. We will write that this work is inspired by observations of Mendenhall Glacier.

35: here Huss & al (2007) should be cited too (add the "e.g.")

Agreed.

38: cite Kessler&Anderson 2004 here too.

Agreed.

68: write "917 kg m^{-3}" and do not use "/". Check the whole manuscript.

We will make this correction throughout.

77: write "Assuming pressurised flow, mass conservation dictates that the rate of change of conduit area is also related to the spatial gradient in discharge,"

We will make this change.

81: write ".. always remains open (Fowler, 1999)."

Okay, we will revise this.

84: I suggest to use another letter than "f" here as "f" is commonly used for the Darcy-Weisbach friction factor, which is the other commonly used discharge relation.

We prefer to keep this as is in order to be consistent with the notation of Kingslake and Ng (2013).

94: state some details of the numerical implementation: spatial discretisation, time stepping algorithm, and maybe other details.

We will expand on some details (line 94), such as spatial discretization and time stepping, but will leave out a full explanation of the numerical methods as it is described in detail in Kingslake (2013) and Kingslake and Ng (2013).

Kingslake, J.: Modelling ice-dammed lake drainage, Ph.D. thesis, University of Sheffield, 2013

Kingslake, J. and Ng, F.: Modelling the coupling of flood discharge with glacier flow during jökulhlaups, Ann. of Glaciol., 54, 25–31, https://doi.org/10.3189/2013AoG63A331, 2013a.

136: state what kind of model it is. I think it is a "higher order" model.

This is a one-dimensional form of the shallow shelf approximation (SSA), which is higher order than the shallow ice approximation (SIA) because it includes membrane stresses. We will expand this sentence to clarify (line 136).

166: state what the spatial discretisation is. Presumably, finite differences? Also state what time-step algorithm is used. A forward Euler step?

Thank you for this comment. You are correct that the model uses finite differences and a forward Euler step. We use a time step of 0.05 yr and an initial grid spacing of 100 m. The model uses a moving grid, so the grid spacing changes with time. We will elaborate on this in the text, but prefer to keep the description short as it is described in detail in Enderlin et al. (2013), which we cite.

178: The terminology around "flotation" is used inconsistently. Here it is stated "ice dam to be at flotation" on line 179 "the basin is not at flotation." This confuses me a bit... another instance is on line 214. More consistent usage would be good.

Thank you for pointing out this inconsistency. We will make revisions to remove instances when we say 'the ice dam is at flotation' and only use the term flotation to refer to the lake or basin depth (Table 1, lines 179, Figure 4, line 214 etc.).

Section 2.4: a table summarising the different simulations would be very helpful.

We agree that summarizing the different simulations would be helpful and we will attempt to address this by adding the flowchart below. The caption for this figure will include the values for the different simulations such Qin, initial water level, and basin geometry.

[Figure]

Fig.4: also include a run here which corresponds to zero floating ice layer thickness. Probably around hw,0=225m

Okay, thank you for this suggestion. Included below is the updated figure with the suggested change.

[Figure]

Fig.5 and other places: I was never quite sure how "storage capacity" was defined (i.e. with or without ice). Maybe "outburst volume" could be used in many instances instead? Either way, define what is used it more prominently than just somewhere in the Results (currently line 218).

We agree that the use of the word 'storage capacity' is confusing. We will define storage capacity earlier (line 50) and at the same time introduce and define "peak water volume". We will also clarify that the peak water volume and basin storage capacity are the same for the flotation scenario, but not for the filling scenario (line 188).

Section 4.2: nice!

Thanks!

316-317: this is definitely not true in general, e.g. Huss et al 2007.

We will change this sentence to say, "One possibility is that in some flood events involving large…".

319: write "flood initiation time" or "flood initiation mechanism" (if that was meant)

Yes, we agree this needs rewording. We will change this to 'flood onset'.

Fig 8a,b not referenced in the text.

Thank you for pointing this out. We will revise the sentences describing Fig. 8c,d to indicate how the flotation scenario differs from the filling scenario.

Fig 8: "Peak floating ice surface" would be clearer (if it fits).

We agree that 'Peak floating ice surface' is what is meant here. We will change 'Peak water surface' to 'Lake highstand' and clarify in the caption that we are plotting the peak water surface level and peak floating ice surface. Also note we will change Fig. 9 to be

consistent with the use of the phrase 'Lake highstand'. Please see both revised figures below.

Fig. 8

[Figure]

Fig. 9

[Figure]

333-334: I had to read this several times to extract the meaning.  Re-word.

We agree that this sentence is overly complicated and will revise this.

354: write "and thus helps"

Okay, we will make this edit.

359: write "the basin drains once it reaches flotation or if it ..."

Agreed.

367-374: Again, I find this list of limitations too prominent for the Conclusions.  If I see a list of "bullet points" in the Conclusions I expect a list of major findings and not short-comings.  Maybe just make the enumeration in-line, thus something like: "In particular, we (1) assumed....; (2) treated ...; (3)...".

Agreed. Thank you for this suggestion.

378: I very appreciate that the code is made available!  It would be great if the README contains a better description, for instance, what scripts need to be run to produce which figures.  Also, ideally, the code is archived into a permanent repository (github is not).  Zenodo does this and readily integrates with github: https://zenodo.org/

We agree the README should contain better description. We have edited the README on Github and published the code in a permanent archive in Zenodo: https://zenodo.org/record/5488047#.YUFKutNKit9

---

## Author Response (AR1)

**Response to Reviewer 1:**

Thank you for your thorough and thoughtful feedback that will help us improve our manuscript.

This manuscript presents a modelling study of the evolution of jökulhlaups from an ice marginal lake as the damming glacier recedes. One novel aspect is (besides the retreating glacier) that it investigates how the floods are affected by remnant ice within the lake basin, due to a tributary glacier disconnecting from the main trunk. The study conducts synthetic experiments with a coupled jökulhlaup and ice dynamics model, both sub-models are of relatively low complexity, which is suitable for the paper's aim, namely, to investigate dynamics of the system in a qualitative fashion. (As an aside: note that fully quantitative modelling of jökulhlaups is not yet possible.)

The topic of the manuscript is interesting and timely, as many new glacier lakes are expected to form under a warming climate. It is well written and is suitable for publication in The Cryosphere once the minor revisions I outline below are incorporated.

We have described how we have addressed each of your comments in our manuscripts in our responses below.

Specific comments

I am a bit confused about the importance of Q_in for the filling scenario. The authors write around line 131 that the choice of Q_in does not impact the results much (which is, by the way, opposite to the finding of Ng & al (2007, doi:10.1029/2007GL031426)). However, then in the Results (line 225) they state "This behavior is sensitive to the model parameters though, as the basin could be made to reach or even exceed flotation by selecting a larger influx Qin." This seems to me like a pretty big impact (and indeed the one I would expect). The authors should state prominently that setting Q_in (the initial channel size S0 has a similar effect) sets how full the lake will get and, consequently, the time and size of the outburst. If the authors feel adventurous, then I would suggest a plot like Fig.6b using a range of S0 (or Qin, although S0 is more arbitrary than Qin, thus I would use that).

We have changed lines 141-146 to read, "In both scenarios, we assume the filling rate Qin remains constant despite the changing climate and year-to-year variability. We tested values of 0-25m^3s^-1 and while different values of Qin impact the flood magnitudes and how quickly a flood is initiated, we found that varying Qin does not change the overall narrative in our results and so we chose to keep Qin constant throughout the filling scenario simulations."

The authors should briefly explain the difference to the Kessler&Anderson (2004, doi:10.1029/2004GL020622)-model (which is what Schoof 2020 uses) who combine a cavity with an R-channel. In particular, that model would avoid having to set an arbitrary initial channel size, albeit, by having to set an almost as arbitrary bedrock bump height. All in all, I think the model used here is totally adequate but still the difference to K&A (2004) should be stated.

We have added that there are models that consider subglacial drainage systems with different configurations such as the ones used in Kessler & Anderson (2004) and Schoof (2020) in line 30 and discuss more generally the theory of outburst floods (lines 28-35).

I think the work on Hidden Creek lake (Kennicott Glacier) should be cited, it is one of the best studied jökulhlaups and also from an ice-marginal lake (albeit, not from a de-glacierized tributary). The generic reference would be Anderson et al (2004, doi:10.1029/2002JF000004). But it might be worth checking whether another of their papers would be more relevant.

We have added the Anderson et al. (2003, doi:10.1029/2002JF000004) reference to line 355.

The authors need to acknowledge that drainage by flotation probably involves other processes than just channel enlargement, the work of Flowers & al (2004, doi:10.1029/2003GL019088) springs to mind.

We have incorporated the citation of Flowers et al. (2004) and acknowledged the potential for drainage to occur through a thin sheet of water (lines 40-42).

Discussion 4.1: here I struggled with this lengthy discussion on something which the model is not capturing, namely ice flow into the basin (lines 255-287). I agree that model limitations should be discussed but starting off the Discussion with this and in particular at this great depth I found a bit much. My suggestion is to move all the mathematics (Eq19-22) and much of the accompanying text into the Appendix and then briefly discuss the limitation in words here and refer to the Appendix for details.

Thank you for this suggestion. We agree this part of the discussion should be rearranged. In order to address this, we moved Eq 19-22 and the relevant accompanying text to Appendix A.

Line by line comments

2: "Marginal basins can form..." as there are other ways that marginal basins form.

Agreed. We have made this change.

Introduction: I think almost all citations need an "e.g." as they are not exhaustive.

Okay. We have added 'e.g.' to relevant citations in the introduction.

19: also cite Werder & al (2010, www.nat-hazards-earth-syst-sci.net/10/227/2010/) here (add the "e.g.")

We have added this reference in line 20 as suggested.

18: there is a new study by Mölg & al (2021, doi:10.1002/esp.5193) for the Swiss Alps which should be cited here.

Thank you for pointing us to this reference. We have added this in line 18.

21: how do proglacial temperatures change rapidly during a flood?  Flood water is as cold as usual proglacial water.

Outburst floods have been observed to cause rapid drops in water temperature in both proglacial rivers and fjords. Figure 11 in Neal (2007) shows water temperatures in the Taku River dropping by about 3 deg C during outburst floods, and Meerhoff et al. (2019) report even larger temperature drops in the Baker River in Patagonia. Kjeldsen et al. (2014) report similar observations for a fjord system in Greenland. We have added the Neal (2007) and Kjeldsen (2014) references in line 21.

Neal (2007): https://pubs.usgs.gov/sir/2007/5027/pdf/sir20075027.pdf

Kjeldsen et al. (2014): https://agupubs.onlinelibrary.wiley.com/doi/10.1002/2013JF003034

26: I'm pretty sure this study was "caused" by the Kienholz & al. (2020) study.  Why not acknowledge that with an extra sentence or two here?

We have added that we are motivated by observations from Mendenhall Glacier to lines 24-25.

35: here Huss & al (2007) should be cited too (add the "e.g.")

We have added this reference to line 43 and "e.g." .

38: cite Kessler&Anderson 2004 here too.

We have added this citation to line 47.

68: write "917 kg m^{-3}" and do not use "/".  Check the whole manuscript.

We have made this correction throughout the manuscript.

77: write "Assuming pressurised flow, mass conservation dictates that the rate of change of conduit area is also related to the spatial gradient in discharge,"

We have made this change.

81: write ".. always remains open (Fowler, 1999)."

Okay, we have added the Fowler (1999) citation.

84: I suggest to use another letter than "f" here as "f" is commonly used for the Darcy-Weisbach friction factor, which is the other commonly used discharge relation.

We prefer to keep this as is in order to be consistent with the notation of Kingslake and Ng (2013).

94: state some details of the numerical implementation: spatial discretisation, time stepping algorithm, and maybe other details.

We created Section 2.1.3 - Numerics, where we expand on some details of the numerical implementation. We leave the full explanation of the numerical methods as it is described in detail in Kingslake (2013) and Kingslake and Ng (2013).

Kingslake, J.: Modelling ice-dammed lake drainage, Ph.D. thesis, University of Sheffield, 2013

Kingslake, J. and Ng, F.: Modelling the coupling of flood discharge with glacier flow during jökulhlaups, Ann. of Glaciol., 54, 25–31, https://doi.org/10.3189/2013AoG63A331, 2013a.

136: state what kind of model it is.  I think it is a "higher order" model.

This is a one-dimensional form of the shallow shelf approximation (SSA), which is higher order than the shallow ice approximation (SIA) because it includes membrane stresses. We have expanded this sentence to clarify (line 165).

166: state what the spatial discretisation is.  Presumably, finite differences?  Also state what time-step algorithm is used.  A forward Euler step?

Thank you for this comment. We have elaborated on this in lines 195-199. This description now says, "The model equations are discretized following the methodology

described in Enderlin et al. (2013) using finite differences for the spatial discretization (initial grid spacing of 100 m), a staggered, moving grid, and a forward Euler time step (Δt=0.05a)." We have kept the description short as it is described in detail in Enderlin et al. (2013), which we cite.

178: The terminology around "flotation" is used inconsistently.  Here it is stated "ice dam to be at flotation" on line 179 "the basin is not at flotation." This confuses me a bit... another instance is on line 214.  More consistent usage would be good.

Thank you for pointing out this inconsistency. We have made revisions throughout the manuscript to replace instances when we say 'the ice dam is at flotation' with 'ice dam flotation' and to replace 'the basin is at flotation' with 'the basin is at flotation depth'.

Section 2.4: a table summarising the different simulations would be very helpful.

We agree that summarizing the different simulations would be helpful and we have attempted to address this by adding the figure (Fig. 5 in the revised manuscript) below. The caption for this figure includes the values for the different simulations such as the discharge into the lake, the initial water level, and the initial cross-sectional area.

[Figure]

Fig.4: also include a run here which corresponds to zero floating ice layer thickness. Probably around hw,0=225m

Okay, thank you for this suggestion. The updated figure below is included in the revised manuscript.

[Figure]

Fig.5 and other places: I was never quite sure how "storage capacity" was defined (i.e. with or without ice). Maybe "outburst volume" could be used in many instances instead? Either way, define what is used it more prominently than just somewhere in the Results (currently line 218).

We agree that the use of the word 'storage capacity' and peak water volume is confusing. We defined basin storage capacity and peak water volume earlier in the text and also clarified that the peak water volume and basin storage capacity are the same for the flotation scenario, but not for the filling scenario (lines 148-151).

Section 4.2: nice!

Thanks!

316-317: this is definitely not true in general, e.g. Huss et al 2007.

We have changed this sentence to say, "For example, in some flood events involving large…" (line 326).

319: write "flood initiation time" or "flood initiation mechanism" (if that was meant)

Yes, we agree this needs rewording. We changed this to 'flood onset' (line 237).

Fig 8a,b not referenced in the text.

Thank you for pointing this out. We have added a reference to this figure (now Fig. 10 in revised manuscript) in line 343.

Fig 8: "Peak floating ice surface" would be clearer (if it fits).

We agree that 'Peak floating ice surface' is what is meant here. We changed 'Peak water surface' to 'Lake highstand' in this figure (now Fig. 10) and clarified in the caption that we are plotting the peak water surface and peak floating ice surface. Also note we changed Figure 11 to be consistent with the use of the phrase 'Lake highstand'. Please see both revised figures below.

Fig. 10

[Figure]

Fig. 11

[Figure]

333-334: I had to read this several times to extract the meaning. Re-word.

We agree that this sentence is overly complicated. We have reworded and broken the sentence into two.

354: write "and thus helps"

Okay, we made this addition.

359: write "the basin drains once it reaches flotation or if it ..."

Agreed. We have made this change.

367-374: Again, I find this list of limitations too prominent for the Conclusions. If I see a list of "bullet points" in the Conclusions I expect a list of major findings and not short-comings. Maybe just make the enumeration in-line, thus something like: "In particular, we (1) assumed....; (2) treated ...; (3)...".

Agreed. Thank you for this suggestion. We have made this change.

378: I very appreciate that the code is made available! It would be great if the README contains a better description, for instance, what scripts need to be run to produce which figures. Also, ideally, the code is archived into a permanent repository (github is not). Zenodo does this and readily integrates with github: https://zenodo.org/

We agree the README should contain better description. We have edited the README on Github and published the code in a permanent archive in Zenodo: https://zenodo.org/record/5488047#.YUFKutNKit9

**Response to Reviewer 2:**

Thank you for your constructive feedback on our manuscript.

In this paper, the authors analyze ice-dammed glacier outburst floods in a model that couples subglacial hydrology to ice flow. Both the subglacial hydrology model and the ice flow model are deliberately simple, yet they still rely on a number of parameters. I really like the approach and, given that coupling hydrology to ice flow will be an important topic in the coming years (cf. D. Brinkerhoff, A. Aschwanden, and M. Fahnestock. Constraining subglacial processes from surface velocity observations using surrogate-based bayesian inference. J. Glaciol., 67(263):385{403, 2021. doi: 10.1017/jog.2020.112.), think that this paper is quite timely. I don't have any strong suggestions, other than a possible citation to the Brinkerhoff paper and that I would like to see plots of the ice evolution, rather than exclusively plots of the peak discharge.

It is not clear to me whether this paper has already gone through one round of revisions and if I am seeing it in the second round of revisions? Nevertheless, this is a great paper, and I think it should be published.

Thank you for the positive comment on the relevance of our manuscript. We would like to clarify that our one-way coupled model only accounts for the impact of ice flow on the outburst floods and subglacial hydrology, but it does not consider the influence of the subglacial hydrology on ice flow. Therefore, we do not believe the Brinkerhoff citation is relevant to our paper as they deal with a fully-coupled model and do not look at short period discharge events over long timescales. We do however, thank you for pointing our attention to the importance of referencing relevant recent work.

We have added two citations to line 55 to place our work into context with other work that has considered how runoff varies as glaciers retreat.

We agree a figure including a plot of ice evolution is helpful. Thank you for this suggestion. To address this, we have added the figure below to the manuscript as Figure 4.

[Figure]